# New Materials Based on Ethylene Propylene Diene Terpolymer and Hemp Fibers Obtained by Green Reactive Processing

**DOI:** 10.3390/ma13092067

**Published:** 2020-04-30

**Authors:** Maria Daniela Stelescu, Elena Manaila, Mihai Georgescu, Mihaela Nituica

**Affiliations:** 1National Research and Development Institute for Textile and Leather, Leather and Footwear Research Institute, 93 Ion Minulescu Street, 031215 Bucharest, Romania; maria.stelescu@icpi.ro (M.D.S.); mihai.georgescu@icpi.ro (M.G.); mihaela.nituica@icpi.ro (M.N.); 2National Research and Development Institute for Lasers, Plasma and Radiation Physics, 409 Atomistilor Street, 077125 Magurele, Romania

**Keywords:** polymeric composite, EPDM, hemp fiber

## Abstract

The paper presents the obtaining of new green polymeric composites using a sustainable reactive processing method, namely electron beam irradiation. EPDM rubber mixtures were reinforced with different amounts of short hemp fibers, which were then irradiated at doses between 75 and 600 kGy. The samples were analyzed by determination of physical–mechanical properties, sol–gel analysis, crosslink density (using the well-known modified Flory–Rehner equation for tetra functional networks), determination of rubber–fiber interactions (using the Kraus equation), water uptake test and FTIR analysis. The obtained results indicate an improvement of the hardness, the tensile and tear strength as the quantity of hemp fibers increases. As the irradiation dose increases, there is an increase in the degree of crosslinking and the gel fraction. Analyzing the behavior of the irradiation samples using the Charlesby–Pinner equation, it is observed that all the samples tend to crosslink by irradiation, the share of degradation reactions being low. For these reasons, the new materials can be used in the food, pharmaceutical or medical field, because the obtained products are sterile and can be easily resterilized by irradiation. They have high elasticity values and can be used to make packaging, seals and other consumer goods.

## 1. Introduction

Ethylene propylene dieneterpolymer (EPDM) rubber is the most widely used synthetic rubber for non-tire applications, such as automotive weather-seals, under-the-hood belts and hoses, roofing membranes, electrical wire and cable, gaskets and many other general rubber industrial goods. EPDM is produced by the terpolymerization of ethene with larger amounts of propylene and a lesser amount of a third monomer with diene structure [1,2]. As with most rubbers, EPDM is always used compounded with fillers to improve its physical–mechanical properties and to reduce its cost. It is possible to use both active fillers that improve the physical–mechanical properties, such as carbon black and silica, as well as inert fillers that lower the cost. Carbon black is obtained via the incomplete combustion of a hydrocarbon feed with natural gas. Currently, silica is obtained by the precipitation method from a silicate salt solution. Others fillers, such as clay, talc and chalk are extracted from the ground and milled to fine powders. These fillers typically used for compounding EPDM, lack sustainability [3].

Numerous studies have been carried out to replace these types of fillers with natural fibers (jute, palm, sisal, hemp, etc.) or other types of organic fillers (wood, cork, soy, etc.) [4,5,6]. Compared to the classic filler, they have many advantages, of which the most important are:(a) low density, high specific strength and stiffness, (b) natural fibers are a renewable resource, (c) natural fibers production requires little energy, (d) natural fibers engage CO_2_ absorption and returns oxygen to the environment, (e) they can be obtained at lower cost compared to typical fillers or synthetic fibers, (f) the production processes are low risk, (g) low emission of toxic fumes when exposed to high temperatures and during incineration at end of life and (h) they are less abrasive compared to the conventional fillers, and thus do not damage the processing equipment. As disadvantages, the following are mentioned: (1) lower durability than composites reinforced with typical fillers due to poor adhesion between the fiber and matrix; it can be improved considerably with treatment, (2) high moisture absorption, which results in swelling, (3) lower strength, in particular impact strength compared with composites reinforced with classical fillers or synthetic fibers, (4) a large variation of the properties and (5) lower processing temperatures limiting matrix options [7,8,9]. 

In order to increase the sustainability of EPDM rubber products, we have explored the technical potential of replacing typical fillers with hemp fibers (Cannabis sativa L) as green alternatives for reinforcing EPDM rubber blends.

The hemp fibers are amongst the cellulose-based natural fibers having the highest physicomechanical properties (density: 1.4–1.5 g/cm^3^, failure strain: 1.5–4%, tensile strength: 310–1110 MPa, specific tensile strength: 210–740 MPa/g cm^−3^) although it should be stated that much variability is seen within the literature [7,10,11]. Hemp is currently the subject of a European Union subsidy for non-food agriculture, and a considerable initiative in currently underway for their further development in Europe [12]. For decades the hemp fibers have been extensively reported as reinforcement of polymer composites in numerous articles and mentioned in most review papers [7,13]. Since these natural fibers are intrinsically hydrophilic, they require surface modification to improve their affinity for hydrophobic polymeric matrices, which enhances the strength, durability, and service lifetime of the resulting natural fiber–polymer composites. Different chemical methods are generally used to modify the surface of natural fibers [14,15]. Recently, research has been leaning instead towards application of ionizing radiation. Ionizing radiation methods are considered superior to chemical methods, due to their advantages: they are clean, require low energy consumption and thus save energy, and are environmentally friendly. Recent applications of irradiation method in obtaining polymeric composites reinforced with natural fibers demonstrated enhanced fiber–polymer interfacial bonding without affecting the inner structure of natural fibers [16].

It is well known that reactions initiated by irradiation with ionizing radiations (electron beam or gamma rays) can be crosslinking, scission or grafting. The crosslinking reactions involve the intermolecular bonds formation between the polymer chains and the crosslinking degree is proportional with irradiation dose. This reaction type does not require unsaturated or other more reactive groups. In contrast, the scission reactions represent the opposite process of crosslinking in which the rupture of C–C bonds occurs. Graft copolymers are obtained by grafting reactions that can occur at low irradiation doses (from 20 kGy) [17].

For these reasons, in this paper, ionizing radiation will be used both for the formation of interfacial bonds between the hemp fibers and EPDM rubber, as well as for the crosslinking of EPDM rubber. In the rubber industry, curing is generally achieved by using sulphur and curing accelerators, occurs at high temperatures and long vulcanization time. It is one of the largest electricity consuming operations, and during the curing, a large number of toxic gases are released (such as nitrosamines–carcinogenic substances). For these reasons, important efforts are being made to replace the classical curing with other more sustainable types of curing. Radiation cross-linking has some advantages over conventional or chemical cross-linking, such as more uniform links formation in the polymer matrix without generating any by-products throughout the decomposition of cross-linking additives, the complete control of cross-linking density, the formation of cross-links in solid state, no need for catalysts or other additives, no heat treatment and the ability of saturated polymers to crosslink, which, because of the absence of unsaturated groups, are largely resistant to conventional chemical treatment. The ionizing radiation process is practically free of waste products and hence there is no serious environmental hazard; the process is very fast, clean and can be controlled precisely [17,18].

EPDM elastomer is categorized as the predominantly radiation-induced cross-linking type of polymers [19]. This property is in accord with the fact that radiation vulcanization of rubbers is one of the promising tools for obtaining valuable products with reasonable physicomechanical properties without any need for adding different ingredients usually used in conventional methods [20]. By processing through electron beam irradiation, in addition to improving properties, the new materials obtained are sterile. The radiation sterilization of polymer materials is an advantageous procedure through which biological contaminants are totally inactivated by the exposure to the action of high energy radiation [21]. Ionizing radiation is highly penetrative and kills bacteria by breaking down bacterial DNA, thereby inhibiting bacterial division. Energy of the ionizing radiation (electron beam or gamma rays) passes through the material, disrupting the pathogens that cause contamination. A minimum dose of 25 kGy was routinely applied for the sterilization of many medical devices, pharmaceutical products and biological tissues [22].

In this paper, a study on radiation crosslinking of EPDM and hemp fibers blending system has been carried out. The obtained mixtures contain different amounts of short hemp fibers, and the irradiation was performed with doses from 75 to 600 kGy. The purpose of the research is to obtain new EPDM rubber materials reinforced with short hemp fibers, in which both the process of improving the interface between natural fibers (which are polar) and the EPDM rubber (which is non-polar), as well as the cross-linking process of the mixtures of rubber in order to change from the plastic to the elastic state, will be made by irradiation with an electron beam (EB). This is a sustainable, ecological process, with a significant reduction in processing time, without release of toxic gases or generating wastes. There are few research papers that address such topics. Most papers in the science literature are related to thermoplastic materials (polyethylene, polypropylene and polylactic acid) reinforced with natural fibers. The numbers of papers on elastomeric materials reinforced with natural fibers are few, some of these papers belong to the authors of this paper [23,24]. In many of the existing papers, different chemical or enzymatic treatments are used to improve the compatibility between natural fibers and polymeric matrix, which involves additional technological operations, additional labor, and generating waste water or other waste [7,15]. At the same time, as it has been shown, by replacing the inorganic fillers, specific to the rubber industry, with organic fillers of the natural fiber type, both the process of obtaining fillers and the rubber products becomes more sustainable, in compliance with the new European requirements regarding the circular economy.

## 2. Materials and Methods

### 2.1. Materials

The following raw materials were used: ethylene-propylene-dieneterpolymers (EPDM) rubber Nordel 4760 produced by Dow Chemical Company, polyethylene glycol PEG 4000 produced by Advance Petrochemicals Ltd., Gujarat, India, antioxidant pentaerythritoltetrakis(3-(3,5-di-tert-butyl-4-hydroxyphenyl) propionate Irganox 1010 from BASF Schweiz, Ciudad de Mexico, Mexico, and the hemp fibers were purchasedas a bundle and were chopped before use, so that thread length would be maximum 2 mm. The main properties of the used materials are presented in Table 1.

### 2.2. Sample Preparation

The compositions of the blends used in this study are shown in Table 2. EPDM rubber blends were prepared on a laboratory electrically heated two-roll mill machine with a cooling system. The working parameters were friction 1:1.1, temperature 80–120 °C and total blending time 10–15 min. The raw materials were added as follows: EPDM rubber was first masticated on the laboratory open two roll mill for 3–5 min, then the antioxidant, polyethylene glycol PEG 4000 were incorporated and finally, the short hemp fibers were included and the blends was homogenized. The thickness of the samples was about 2 mm thick.

Plates required for tests (sheets of 300 × 300 × 2 mm^3^) have been made by compression molding, using a laboratory electrical press Fortune Presses model no. TP 600 manufactured by Fontijne Grotnes, Vlaardingen, The Netherlands, at a temperature of 160 °C, a pressing force of 300 kN and a time of 5 min. The plates were then cooled to room temperature under a pressing force of 300 kN.

### 2.3. Experimental Installations and Sample Irradiation

The samples were irradiated using the electron beam accelerator called ALIN 10 in the dose range of 75–600 kGy. The description of the electron beam accelerator and the irradiation method are presented in our previous studies [17,19,23,24].

### 2.4. Laboratory Tests

#### 2.4.1. Mechanical Characteristics

The following mechanical properties have been determined: tensile strength, tearing strength, elongation at break, residual elongation, hardness and elasticity. The determination methods are in accordance with ISO 37/2017, ISO 7619-1/2011 and ISO 4662/2017.

#### 2.4.2. Sol–Gel Analysis

The measurement of the fraction of rubber remaining unreacted to the network as a function of the degree of the curve has been used to estimate the extent of chain scission in the EPDM/hemp fiber samples crosslinked by EB. The determination method is presented in previous studies [17,19,23,24]. The gel fraction was calculated as follows:(1)$$\mathrm{Gel}\;\mathrm{fraction}\;(\%)=\frac{m_{s}}{m_{i}}\times 100$$
where *m_s_* and *m_i_* are the weight of the dried sample after extraction and the weight of the sample before extraction, respectively [19,26]. 

#### 2.4.3. The Crosslinking Density (ν)

The crosslinking density (ν) of the samples was determined on the basis of equilibrium solvent-swelling measurements (in toluene at 23–25 °C) by application of the well-known modified Flory–Rehner equation for tetra functional networks and the method is presented in previous studies [17,19,23,24]. The volume fractions of polymer in the samples at equilibrium swelling *ν*_2m_ were determined from swelling ratio G as follows:(2)$$\nu_{2m}=\frac{1}{1+G}$$
where:(3)$$G=\frac{m_{g}-m_{s}}{m_{s}}\times \frac{\rho_{r}}{\rho_{s}}$$
where *m_s_* and *m_g_* are the weight of the dried sample after extraction and the weight of the swollen sample, respectively and $\rho_{r}$ and $\rho_{s}$ are the densities of rubber samples and solvent. The densities of elastomer samples were determined byhydrostaticweighingmethod, according to the ISO 2781/2018. The crosslink densities of the samples, *ν*, were determined from measurements in a solvent, using the Flory–Rehner relationship:(4)$$\nu =-\frac{Ln\left(1-\nu_{2m}\right)+\nu_{2m}+\chi_{12}\nu_{2m}^{2}}{V_{1}\left(\nu_{2m}^{1/3}-\frac{\nu_{2m}}{2}\right)}$$
where $V_{1}$ is the molar volume of solvent (106.52 cm^3^/mol for toluene), $\nu_{2m}$ is the volume fraction of polymer in the sample at equilibrium swelling, and $\chi_{12}$ is the Flory–Huggins polymer–solvent interaction term (0.49 for EPDM-toluene) [19,26,27].

#### 2.4.4. Rubber–Fiber Interactions

The extent of interaction between rubber and fiber can be analyzed using Kraus equation [28,29]. The Kraus equation is as follows:(5)$$V_{ro}/V_{rf}=1-m\frac{f}{1-f}$$
where *f* is the volume fraction of fiber, *m* is the fiber polymer interaction parameter and *V_ro_* and *V_rf_* are the volume fractions of rubber in the EPDM and EPDM/fiber swollen sample, respectively. *V_rf_* was calculated by the expression:(6)$$V_{rf}=\frac{\left[(D-FT)/\rho_{r}\right]}{\left[(D-FT)/\rho_{r}]+[A_{0}/\rho_{s}\right]}$$
where $\rho_{r}$ and $\rho_{s}$ are the densities of EPDM rubber and solventrespectively, *D* is the deswollen weight of the test specimen (dry weight), *F* is the weight fraction of the insoluble components, *T* is the initial weight of the specimen and *A*_0_ is the weight of the absorbed solvent at swelling equilibrium. For a sample without filler, *F* = 0 and the expression for the volume fraction of rubber in the solvent-swollen unfilled vulcanizates (*V_ro_*) is obtained using Equation (6).

#### 2.4.5. Water Uptake Test

The effect of water absorption (at room temperature 23 ± 2 °C) on hemp fiber reinforced EPDM rubber composites are investigated in accordance with ISO 20344/2011. The water uptake was calculated using the following Equation:(7)$$\mathrm{Water}\;\mathrm{uptake}\;(\%)=\frac{m_{S}-m_{1}}{m_{1}}\times 100$$
where *m_S_* is the weight of the sample saturated with water, determined at periodic intervals and *m*_1_ is the initial weight of the oven-dried sample [30].

#### 2.4.6. Fourier Transform Infrared Spectroscopy (FTIR)Analysis

The structure of the EPDM-hemp fibers composites cross-linked by EB irradiationwas analyzed by FTIR measurements using Nicolet iS50 FT-IR spectrophotometer. The absorption spectra were obtained as an average of 20 scans, in the range of 1800–450 cm^−1^, with a resolution of 4 cm^−1^. 

## 3. Results and Discussion

### 3.1. Mechanical Properties

The mechanical properties of composites based on EPDM rubber and hemp fibers are presented in Figure 1, Figure 2, Figure 3, Figure 4, Figure 5 and Figure 6. It can be observed that the values for tensile strength (Figure 1) and tearing strength (Figure 2) for mixtures without fillers, increase with the increase of the irradiation dose. They reach an optimum value and afterwards they decrease. This behavior can be interpreted by the theory of amorphous elastomers, that is, when the dose of EB increases as a result of the crosslinking reactions that occur, an increase of the crosslinking density takes place. The tensile strength and energy stored in an amorphous elastomer upon rupture is closely related to the chain deformation and only very little to the surface energy that resulted from breaking. The part of the effective elasticity of the network chains increases with the initial molecular mass and with the crosslinking density, while the stretch decreases with the increase of the crosslinking density. From this it is obvious that the resistance can only become maximum through a proper balance between the effective elastic part of the network chains, and the extensibility of the network. Thus, both the tensile strength and the stored energy will be maximum when all the chains have been stretched to the maximum value, that is, they are oriented so that they represent a continuous bundle of aligned filaments [31]. In the mixtures shown, the behavior is influenced both by the amount of short hemp fibers introduced into the mixture, as well as by the fact that the EPDM elastomer used has a high ethylene content and is crystalline (10% crystallinity). The crystallization that can take place acts as a false crosslinking, the rubber composition becoming rigid. The degree of crystallinity decreases with the introduction of organic filler (hemp fibers), as well as by irradiation [23]. Therefore, when interpreting the data presented in Figure 1, Figure 2, Figure 3, Figure 4, Figure 5 and Figure 6 by the theory of amorphous elastomers, it should be taken into account that certain variations in the properties of the samples may be associated with crystallization phenomena. For the 75 kGy dose, the increase of the two characteristics is observed with the increase of the hemp fibers amount, an increase similar to the one observed by the use of active fillers [32], showing a good compatibility between elastomer and natural fibers [23,33].

Elongation at breakvalues (Figure 3), for a radiation dose of 75 kGy, increase with the increased amount of fibers introduced, due to the strong rubber–filler interaction that occurred in the system. For the mix with 20 phr hemp fibers, a smaller value is observed for elongation at break. Given that for this sample the best values of hardness, tensile and tear strength were obtained, it can be assumed that this value of the elongation at break would be due to the enhancement in rigidity of the composites [34] or a weaker rubber–fiber interaction. Upon increase of the irradiation dose, the value of the elongation at break decreased, as a result of the increase of the degree of crosslinking that limits the movement of the macromolecular chains.

Analyzing the values of the residual elongation (Figure 4) and comparing with elongation at break or elasticity, it was observed that the best return to the initial form had the mixture without fillers, and as the amount of hemp fibers increased, the return to the original form was difficult. When increasing the dose of irradiation there was a decrease in the residual elongation of the samples, due to the increase of the crosslinking degree, which helped to return to the initial form after the chains were stretched.Hardness (Figure 5) increased as the amount of hemp fibers increases showing that they had an effect of reinforcing the EPDM rubber mixtures, similar to the active fillers. When the irradiation dose is increased, the hardness does not vary significantly [17].

The elasticity (Figure 6) decreased as the amount of high elasticity amorphous elastomer (EPDM) decreased and the amount of hemp fibers increased, which reduced elasticity and segment mobility of the EPDM [35].

### 3.2. Gel Fraction 

The gel content of the EPDM/hemp fibers composites (Figure 7) increased with increasing irradiation dosage, the highest values were obtained for samples irradiated with 600 kGy. A maximum gel content of 96.22% was obtained for the sample with 15 phr of hemp fibers, irradiated with 600 kGy, and the minimum of 89.16% was obtained for the sample with 15 phr of hemp fibers, irradiated with 75 kGy. The obtained results confirm the tendency of EPDM to crosslink by irradiation [17,19].

### 3.3. Crosslink Density

In Figure 8 the crosslink density values are shown, determined using the Flory–Rehner equation. For the analyzed samples, an increase of the crosslinking density occurred with the increase of the EB dose, but when thehemp fiber content increased, an uneven variation of the crosslinking density was observed. These results were due, in the first place, to the reactions that occur through irradiation, namely: *(1)* the crosslinking reactions that occurred especially in the elastomeric matrix of EPDM, *(2)* the degradation reactions that occurred especially in the hemp fibers and which could lead to scission of the main chain and the formation of radicals, which may initiate grafting, crosslinking reactions, or obtaining aldehydes, and carboxylic acids [36] and *(3)* grafting and recombination reactions, which can lead to the improvement of rubber–fibers interactions, as well as the structural changes of the samples [17,19,23].

In order to quantitatively evaluate the crosslinking and chain scission yields of the composites based on EPDM and hemp fiber, the Charlesby–Pinner equation can be used [37,38]:(8)$$S+\sqrt{S}=\frac{p_{0}}{q_{0}}+\frac{1}{\alpha P_{n}D}$$
where, *S* is the sol fraction *(S = 1-gel fraction)*; *p*_0_ is the fracture density per unit dose; *q*_0_ is the density of crosslinked units per unit dose; *P_n_* the number averaged degree of polymerization and *D* is the radiation dose in Gy.

Figure 9 shows the graphical representation of the Charlesby–Pinner equation for all samples, and the results obtained regarding the ratio between the degradation density (*p*_0_) and the crosslinking density (*q*_0_) are displayed in Table 3. The obtained results indicate that when increasing the amount of hemp, the share of the crosslinking reactions was 3.8–4.5 times higher than that of the degradation reactions. Except for the mixture that contained 5 phr hemp fibers (which had a ratio value *p*_0_/*q*_0_ lower than the control sample), for the samples with hemp fibers a slight increase in the *p*_0_/*q*_0_ ratiocould be observed, which may be due to the tendency of hemp fibers to degrade by irradiation [36].

### 3.4. Rubber–Fiber Interactions

In order to determine rubber–fiber interactions, the Kraus equation was used from swelling time [28,29]. The ratio *V_ro_/V_rf_*, and the values *V_rf_* are listed in Table 4. The fiber polymer interaction parameter (*m*) was determined by graphically representing the Kraus equation (see relation 6) and the obtained values are shown in Figure 10, where in the *option a* theparameter *m* was determined for all four hemp fiber concentrations, and in *option b*, the parameter *m* wasdetermined only for the first three concentrations of fibers, removing the 20 phr hemp fibers sample. According to Kraus theory [28], the negative values of the parameter *m*, suggesting an improved interfacial interaction. For the first option, negative values were obtained only for the samples irradiated with 75 kGy, on the other hand, if only the samples with 5 phr, 10 phr and 15 phr hemp fibers were analyzed, negative values of the parameter *m* were obtained for irradiation doses of 75 kGy, 150 kGy and 300 kGy, respectively. In conclusion, for samples with a concentration of hemp fiber of max 15 phr irradiated with 75–300 kGy, as well as for mixtures with 20 phr hemp fiber irradiated with 75 kGy, very good values of the rubber–fiber interaction were obtained. These experimental data indicate that the new method of improving rubber–fiber interaction by accelerated electron irradiation leads to great results. 

### 3.5. Water Uptake 

The water uptake results obtained for the EPDM-based composites and hemp fibers are shown in Figure 11. The percentage of water absorption in these samples depends on both the hemp fibers content and the irradiation dose. The water uptake for all samples was faster during the first 600 h (h) and continued to increase progressively, when finally, a plateau was observed, showing that the water absorption reached equilibrium (Figure 12). For composites containing 5 phr and 10 phr hemp fibers, a significant increase in water absorption was observed (approximately 3, respectively 5 times compared to the control sample), but the further increase of the amount of hemp fibers in mixtures, the changes that take place in the value of water absorption are uneven. The increase of water absorption is due to the hydrophilic nature of the hemp fiber, which increased the absorption capacity of the water by forming hydrogen bonds between the water and the cellulose and hemicellulose -OH groups. Thus, with the increase of the hemp content in the composite, the number of hydrogen bonds between the organic components and the water molecules also increased. Higher hemp content led to larger gaps that led to water accumulation at the interface between the fiber and matrix [39,40]. The degree of water absorption depends on the availability of –OH groups on the fiber surface. In composites with a high degree of compatibilization, the number of –OH groups was reduced by the irradiation process (observed at 75, 150 and 300 kGy) and thus the water absorption was restricted. Moreover, the gaps in the composites with a higher degree of compatibility were reduced due to the improved adherence to the interface between the fiber and the matrix, which has limited the penetration of water molecules into composites [39].When increasing the dose of irradiation, a slight increase in the equilibrium value of the water absorption was observed for the doses of 300 and 600 kGy only for mixtures containing 15 phr and 20 phr, respectively. This can be determined by the changes in the structure of the composites by irradiation at high doses, which led to both the modification of the sample surface and the fiber–rubber interface.

### 3.6. Fourier Transform Infrared Spectroscopy (FTIR) Analysis

For evaluating the structural modification due to processing by the irradiation of the samples, analysis of the spectral FT-IR of the samples was carried out. Figure 13 and Figure 14 present the infrared spectra obtained in the range of 1800–450 cm^−1^ of the blends irradiated, without fiber hemp and 20 phr of fiber hemp.

In Figure 13 the specific bands observed by irradiation crosslinked EPDM rubber, among which we mention [41,42]: (a) the bands from 1465 cm^−^^1^ and 1376 cm^−1^ corresponding to the shear (breaking) vibrations of the -CH_2_– group, respectively of the bending of the C-H group in the -CH_3_ group. (b) The band located at 720 cm^−1^,which is a typical balancing vibration of the -(CH_2_)_n_- groups type when *n* ≥ 5 and it is in our case, due to the grouping -CH_2_- of the EPDM rubber chains.

For the sample with 20 phr natural fibers irradiated at the same dose of 150 kGy, a decrease in intensity of 1465 cm^−1^ band was observed, specific shear vibrations (breaking) of the group -CH_2_, and respectively of the band from 1376 cm^−1^ specific to the CH group from the -CH_3_ groups. These bandswere overlapped over the bands attributed to C–H bending as well as -CH_2_ rocking vibration of groups in lignin and cellulose [43,44,45]. In both cases a sharp band was observed at the 1760–1740 cm^−1^ that might be assigned to C = O stretching of the carboxylic group of hemicellulose [38], or the formation of-COOH groups during the oxidation/degradation of EPDM by irradiation in the atmospheric conditions [42,43].

For samples containing hemp fibers, there were several bands that differentiate from existing ones at EPDM, such as [43]: (1) at 650–700 cm^−1^ we observed the specific bond stretching band of –OH bending out of phase, existing in cellulose or hemicelluloses; (2) two bands located between 1000 and 1100 cm^−1^ specific C-O stretching bond, skeletal vibration CO and nonsymmetric in-phase ring; (3) the presence of a band at 1330–1340 cm^−1^ specific to S ring stretching of cellulose; (4) compared to the fiber-free mixture, in the mixes reinforced with hemp fibers, a movement of the bands from 730 to 900 cm^−1^ was observed (of which the one from 872 cm^−1^ is highlighted), specific to the mono, di- and tri- substituted bending CH bonds, and the appearance of other specific bands of cellulose, hemicellulose or lignin that are part of the hemp fibers that appear at 1000–1100 cm^−1^, these being due to CC, C-OH, CH ring and side group vibrations.

Analyzing Figure 14, it was observed that for the mixing with 20 phr hemp fibers, when increasing the irradiation dose from 75 to 600 kGy, all the bands presented above were maintained, but revealed variations in the intensities of specific bands, such as: (a) in the region 1115–1370 cm^−1^, the bands with the highest intensity are those of the sample irradiated with 150 kGy; as specified in this section there is a band at 1330–1340 cm^−1^ specific to the S ring stretching of the cellulose, and the bands of specific cellulose such as: C-O-C symmetric stretching, OH plane deformation and C-OH bending at C6 [43]. (b) In the region 950–1110 cm^−1^ and 500–720 cm^−1^, the most intense are the bands of the irradiated sample with 300 kGy, followed by the sample with 150 kGy, 600 kGy and then the one with 75 kGy. In theseregion, we observed the specific bands of cellulose, hemicellulose or lignin that are part of the hemp fibers, such as: the specific bond stretching bond of –OH out of phase bending existing in cellulose or hemicellulose (650–700 cm^−1^), two specific bond stretching CO bands, skeletal vibration C-O and nonsymmetric in-phase ring (1000–1100 cm^−1^) respectively.

The bands attribution in samples from the FTIR spectra presented above are listed in Table 5.

From the data presented it could be concluded that when increasing the irradiation dose there were changes in the structure of the composites, but there were no significant changes that could indicate a degradation of the samples. It can be observed that a dose of 75 kGy was sufficient to consume the double C = C bonds existing in EPDM (the specific band had to appear at 1620–1640 cm^−1^) [46]. The obtained results are in agreement with the presented physicomechanical properties, which indicate a good behavior of the samples at high irradiation doses, as well as with the determinations regarding the cross-linking density, which shows that as the irradiation dose increases further, changes in samples structure occur, as a result of crosslinking reactions leading to increased crosslinking.

## 4. Conclusions

New types of green composites based on EPDM rubber and hemp fibers were obtained. The chemical methods of modifying the fiber surface in order to improve the fiber–rubber interfacial bonding, as well as the classical crosslinking of the elastomeric matrix, have been replaced with electron beam irradiation—a sustainable reactive processing method for polymeric materials. Ashemp fiber content increasedto 20 phr, an improvement in hardness, tensile and tear strength was observed. Both gel fraction and crosslink density values or FTIR analyses indicate efficient crosslinking even at a dose of 75 kGy. For samples with a hemp fiber concentration of 15 phr irradiated with 75–300 kGy, as well as blends with 20 phr of hemp fibers irradiated with 75 kGy very good values of the rubber–fiber interactions, indicating that the new method of irradiation with EB led to an improvement of the rubber–fiber interaction. The equilibrium water uptake values (after 1200 h) were below 12% for all samples, a good value, which allowed the use of new materials in various domains. The obtained results regarding the ratio between the degradation density (p_0_) and the crosslinking density (q_0_) using the Charlesby–Pinner equation, show that for the composites made, the share of the cross-linking reactions was 3.8–4.5 times higher than the degradation reactions. These aspects were also confirmed by the FTIR analyses. By processing through electron beam irradiation, besides the mentioned properties, the new materials obtained were sterile (one of the fastest, most efficient and sustainable methods of sterilization was irradiation with doses of min 25 kGy). Good behavior to irradiation of the developed materials allows finished products made thereof to be repeatedly resterilized. This property greatly extended the range of applicability, the new materials can be used successfully in the food, pharmaceutical or medical field, for the manufacture of packaging, sealing materials, etc.

## Figures and Tables

**Figure 1 materials-13-02067-f001:**
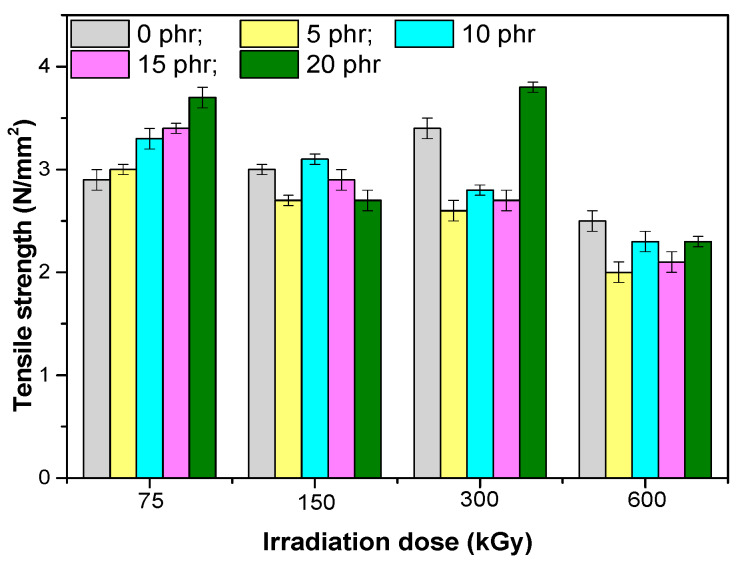
Tensile strength versus electron beam irradiation dose and the amount of hemp fibers.

**Figure 2 materials-13-02067-f002:**
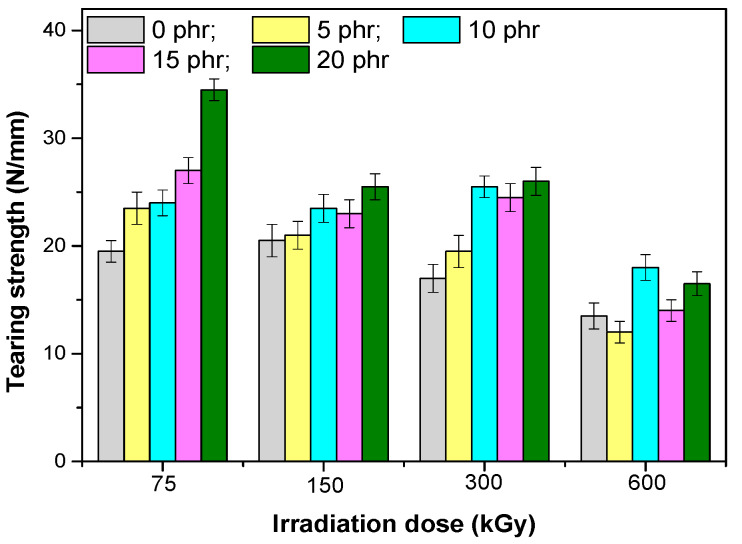
Tearing strength versus electron beam irradiation dose and the amount of hemp fibers.

**Figure 3 materials-13-02067-f003:**
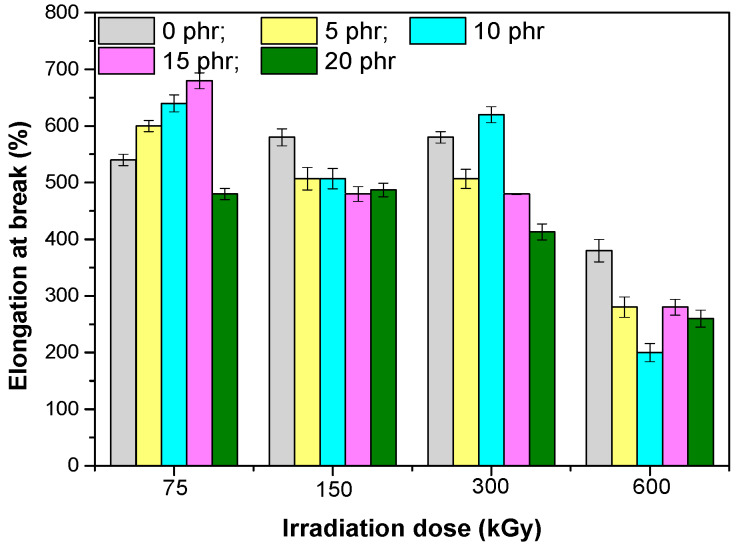
Elongation at break versus EB irradiation dose and the amount of hemp fibers.

**Figure 4 materials-13-02067-f004:**
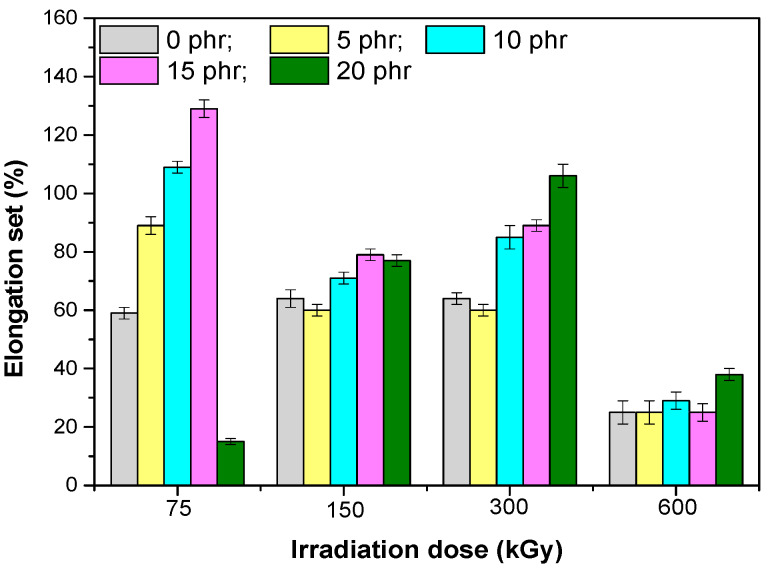
Residual elongation versus EB irradiation dose and the amount of hemp fibers.

**Figure 5 materials-13-02067-f005:**
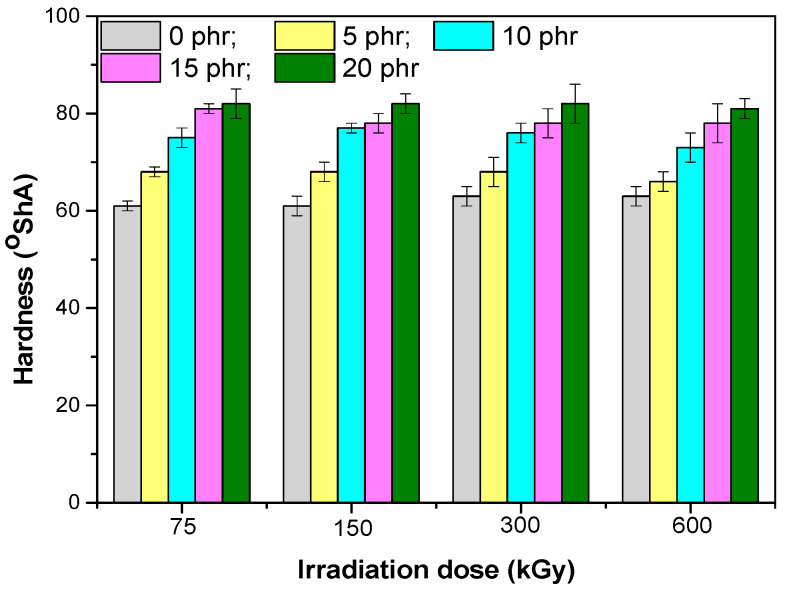
Hardness versus electron beam irradiation dose and the amount of hemp fibers.

**Figure 6 materials-13-02067-f006:**
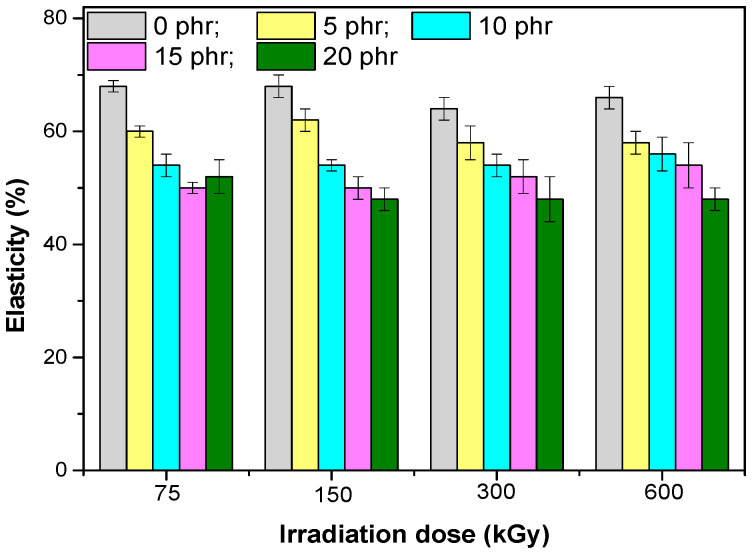
Elasticity versus electron beam irradiation dose and the amount of hemp fibers.

**Figure 7 materials-13-02067-f007:**
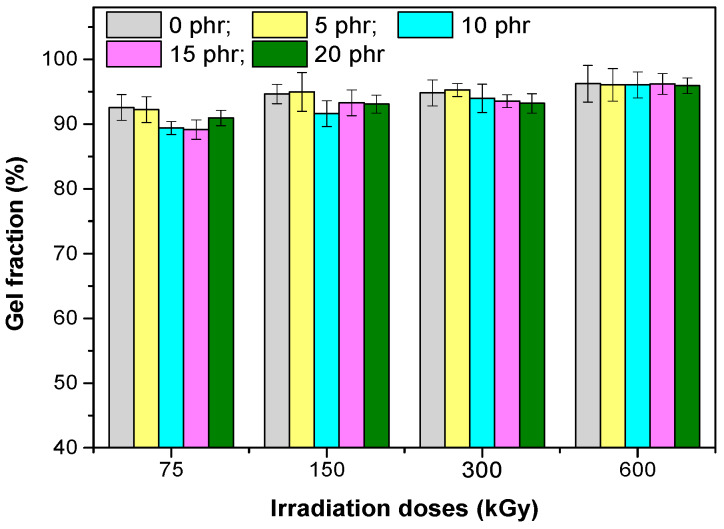
Gel fraction versus electron beam irradiation dose and the amount of hemp fibers.

**Figure 8 materials-13-02067-f008:**
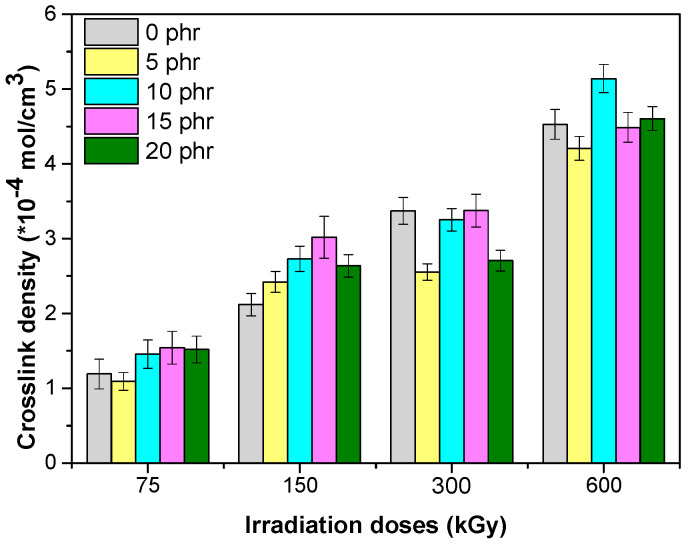
Crosslink density versus electron beam irradiation dose and the amount of hemp fibers.

**Figure 9 materials-13-02067-f009:**
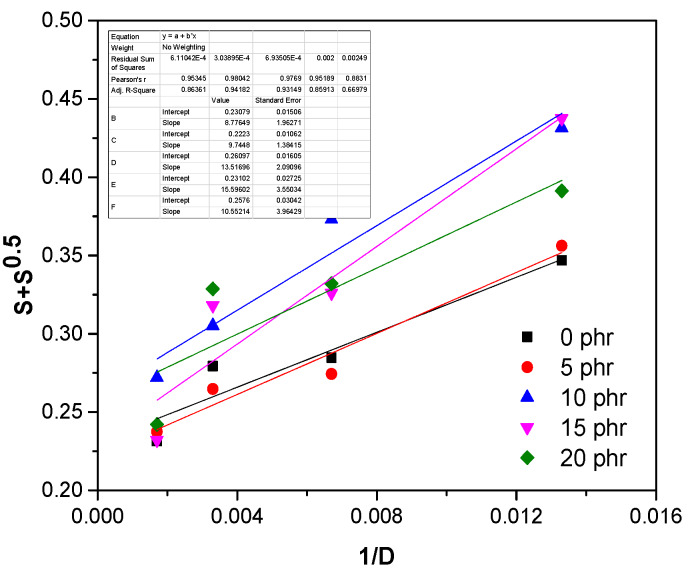
Charlesby–Pinner plots of EPDM/hemp samples.

**Figure 10 materials-13-02067-f010:**
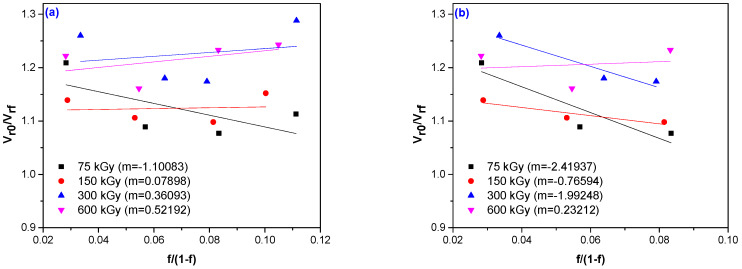
Graphical representation of the Kraus equation for EPDM/hemp fiber composites at different irradiation doses in order to determine the values of parameter ‘*m*’ for: (**a**) mixtures with 5, 10, 15 and 20 phr hemp and (**b**) for mixtures with 5, 10 and 15 phr hemp

**Figure 11 materials-13-02067-f011:**
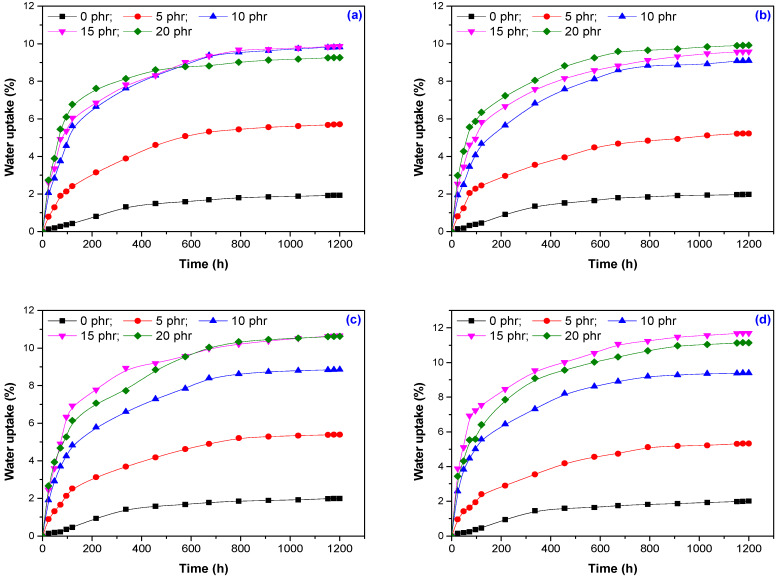
Water uptake of composites based on EPDM and hemp fiber versus immersion time for irradiation doses of (**a**) 75 kGy; (**b**) 150 kGy; (**c**) 300 kGy and (**d**) 600 kGy.

**Figure 12 materials-13-02067-f012:**
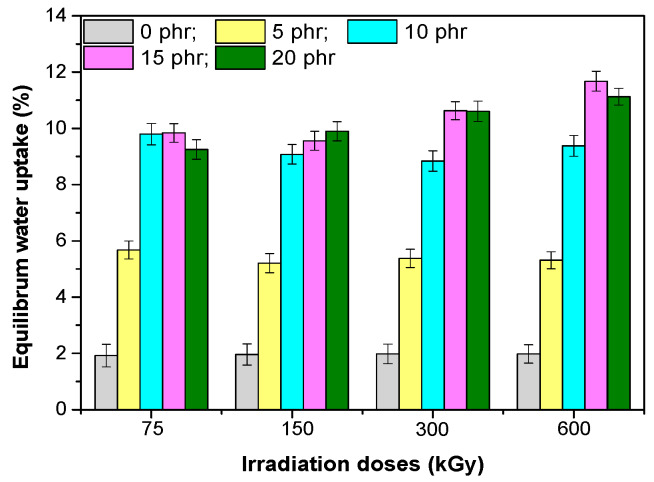
Equilibrium water uptake in depending on the dose of irradiation for composites based on EPDM and hemp fiber.

**Figure 13 materials-13-02067-f013:**
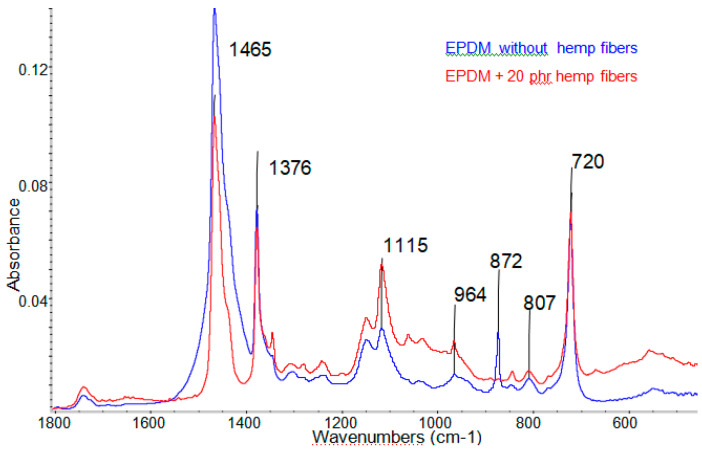
The infrared spectra for EPDM samples without hemp fibers and with 20 phr hemp fibers, irradiated with 150 kGy.

**Figure 14 materials-13-02067-f014:**
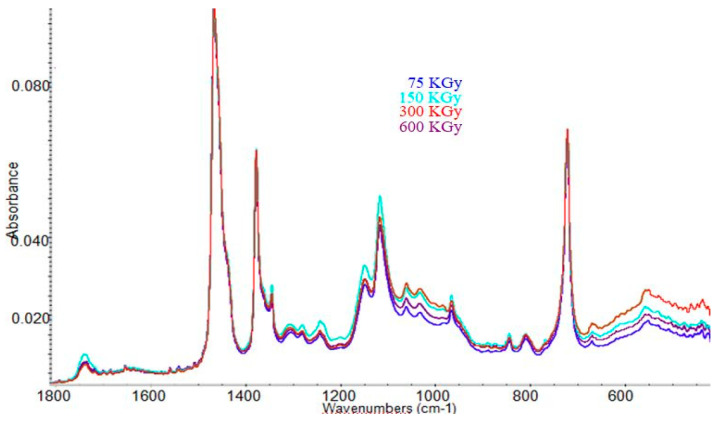
The infrared spectra for EPDM samples with 20 phr hemp fibers irradiated at different doses (from 75 to 600 kGy).

**Table 1 materials-13-02067-t001:** The main properties of the materials used.

The Materials	Functions	Properties
Ethylene-propylene-dieneterpolymersNordel 4760	Acts as a matrix	Mooney viscosity is 70 ML_1+4_ at 120 °C, 70% ethylene content, 5-ethylidenenorbornene (ENB) 4.9 wt %, density 0.88 g/cm^3^, 10% crystalline degree
The hemp fiber	Acts as a filler	Thread length of max 2 mmComposition: 68–92 wt % cellulose, 15–22 wt % hemicellulose, 3–10% lignin, 0.8 wt % wax, the rest are fat, pectin [10,11,12,25].
Polyethylene glycolPEG 4000	Acts as a lubricant and a plasticizer	1.128 g/cm^3^ density, 4–8 °C melting point range
Irganox 1010	Acts as an antioxidant	Melting point of 40 °C, 98% active ingredient

**Table 2 materials-13-02067-t002:** Compound formulation.

Ingredients	Loading in phr *
EPDM rubber	100
Hemp fibers	0–20
PEG 4000	3
Antioxidant	1

* phr—parts per hundred parts rubber.

**Table 3 materials-13-02067-t003:** Compositional characteristics, designation and p_0_/q_0_ ratio for EPDM and EPDM/hemp samples.

Samples	p_0_/q_0_
EPDM	0.2308
EPDM+5 phr hemp	0.2223
EPDM+10 phr hemp	0.2610
EPDM+15 phr hemp	0.2310
EPDM+20 phr hemp	0.2576

**Table 4 materials-13-02067-t004:** Vrf and *Vro/Vrf* values of composites based on EPDM and flax fiber depending on the amount of hemp fibers and the irradiation dose.

Irradiation Doses (kGy)	The Amount of Hemp Fibers							
	*5 phr*		*10 phr*		*15 phr*		*20 phr*	
	*^V^_rf_*	*^V^_ro_/^V^_rf_*	*^V^_rf_*	*^V^_ro_/^V^_rf_*	*^V^_rf_*	*^V^_ro_/^V^_rf_*	*^V^* *_rf_*	*^V^_ro_/^V^_rf_*
75 kGy	0.229	1.209	0.254	1.089	0.257	1.077	0.248	1.113
150 kGy	0.305	1.139	0.314	1.106	0.316	1.098	0.301	1.152
300 kGy	0.309	1.260	0.330	1.180	0.332	1.174	0.302	1.243
600 kGy	0.362	1.222	0.381	1.161	0.359	1.233	0.356	1.288

**Table 5 materials-13-02067-t005:** Main spectral attributions in EPDM/hemp fibers composites.

Band Position in the EPDM-Hemp Composites (cm^−1^)	Functional Group
1740–1760	C=O stretch in non-conjugated ketones, carbonyls and in ester groups
1465	CH_2_ bending and rocking vibrations from EPDM
1376	CH_3_ bending vibration from EPDM
1330–1340	S ring stretching from cellulose
1115–1330	C-O-C symmetrical and asymmetrical stretching, OH plane deformation, C-OH bending at C6
1000–1100	bond stretching C-O, skeletal vibration C-O, non-symmetric in-phase ring, respectively
730–900	mono-, di- and tri-substituted C-H bending
720	CH_2_ bending and rocking vibrations from EPDM
650–700	bond stretching of –OH out of phase bending from cellulose

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
