# Peer review of "New Materials Based on Ethylene Propylene Diene Terpolymer and Hemp Fibers Obtained by Green Reactive Processing"

_materials, 2020, doi:10.3390/ma13092067_

Round 1

Reviewer 1 Report

Interesting work on hemp fibers introduced into a rubbery matrix, and crosslinked by irradiation.

Only some small points to revise, before approval:

You mention sterilization potential by beam irradiation. This is central to the scope of the work, and the fact that FTIR and mechanical properties don’t show substantial degradation even with high beam irradiation, you should add a few lines at the end of the Introduction, stating that your work does also investigate possible sterilization and quoting one or two references about sterilization by beam irradiation and relevant levels of irradiation needed.

Please indicate where you quoted from Reference 12, since you didn’t mention it.

Please mention where you get data from Table 1 about hemp (e.g., the supplier, or did you measure it? Some of the data appear of interest, since not very typical, like the 68% of cellulose)

Which antioxidant did you use? Please clarify,

Author Response

  1. I have introduced a paragraph in the chapter "Introduction" with reference to irradiation sterilization:

„By processing through electron beam irradiation, in addition to improving properties, the new materials obtained are sterile. The radiation sterilization of polymer materials is an advantageous procedure through which biological contaminants are totally inactivated by the exposure to the action of high energy radiation [21]. Ionizing radiation is highly penetrative and kills bacteria by breaking down bacterial DNA, thereby inhibiting bacterial division. Energy of the ionizing radiation (electron beam or gamma rays) passes through the material, disrupting the pathogens that cause contamination. A minimum dose of 25 kGy was routinely applied for the sterilization of many medical devices, pharmaceutical products and biological tissues [22].”

[21] T. Zaharescu, S. Jipa, B. Gigante, „Stabilized polyethylene on the sterilization dose range”, Polymer Bulletin 57, 729–735 (2006); DOI 10.1007/s00289-006-0628-x  

[22] Rodrigo Navarro, Guillermina Burillo, Esbaide Adem, Angel Marcos-Fernández, „Effect of Ionizing Radiation on the Chemical Structure and the Physical Properties of Polycaprolactones of Different Molecular Weight”, Polymers, 2018, 10(4), 397; DOI:10.3390/polym10040397

 2.  It is our mistake. The paragraph " Hemp is currently the subject of a European Union subsidy for non-food agriculture, and a considerable initiative in currently underway for their further development in Europe [2]" has reference [12], not [2]

3. Data on the chemical composition of hemp were taken from the literature. We modified that paragraph and added two more bibliographic references.

Thus, the paragraph " Composition: 68 wt% cellulose, 15 wt% hemicellulose, 10 % lignin, 0.8 wt% wax, the rest are fat, pectin [13]." was modified with the pragraph "Composition: 67-92 wt% cellulose, 11-22 wt% hemicellulose, 3-10 % lignin, 0.8-0.9 wt% wax, the rest are fat, pectin [10-12, 25]".

 [10]. J. P. Manaia, A. T. Manaia, L. Rodriges, Industrial Hemp Fibers: An Overview, Fibers, 2019, Volume 7, 106; doi:10.3390/

[11]. L.Y. Mwaikambo, Review of history, properties, and application of plant fibres, Afr. J. Sci. Technol. 7 (2006) 120–133.

[12]. O. Faruk, A.K. Bledzki, H.-P. Fink, M. Sain, Biocomposites reinforced with natural fibers: 2000–2010, Prog Polym Sci , 2012, Volume 37, pp. 1552– 1596.

[25]. Shahzad A., Hemp fiber and its composites - A review, Journal of Composite Materials , 2012, Volume 46(8), pp. 973-986, DOI: 10.1177/0021998311413623

4. The antioxidant used in the samples preparation was Irganox 1010 - pentaerythritol tetrakis (3-(3,5-di-tert-butyl-4-hydroxyphenyl).

Reviewer 2 Report

The manuscript by Stelescu et al. reports a study on the EPDM rubber composites with hemp fibers modified by the electron beam irradiation. The reviewed work is interesting and worth of publishing, especially due to the green production technique and high application potential. Despite of a potential interest for readers, the manuscript still suffers from some shortcomings and nomenclature errors hindering publications of their results in the present form.

Reviewer suggestion

For better consistency and clarity of the article, I suggest presenting the results in the following order: FTIR analysis, crosslinking density, sol-gel fraction analysis, water uptake test, rubber-fiber interactions, mechanical properties. The observed changes in mechanical behavior are results of all other structural changes observed for the described mixtures.

Minor comments:

  1. Sentence started in line 43 and 375 need to be rewrite more clearly.
  2. Nomenclature or language errors:
  • In line 86 should be “groups” instead of “groupings”
  • In line 176 should be “hardness” instead of “hardner”
  • In line 180 should be “unreacted” instead of “unconnected”
  • In line 189 should be “crosslinking density” instead of “crosslink densities” – please correct it throughout
  • In line 299 should be “hemp fiber content increases” instead of “hemp fiber increases” – the same in line 433
  • On figure 14 – in legend are Mrad units whereas in the text Authors used kGy, please uniform this
  • In lines 387, 397 should be “bands/band” instead of “strips/strip”
  • In lines 387, 397 should be “bands/band” instead of “strips/strip”

Please change word “section” to “region” in the FTIR part e.g. line 405 and further

Author Response

  1. Thank you for your suggestion regarding the presentation of the results in another order (FTIR analysis, crosslinking density, sol-gel fraction analysis, water uptake test, rubber-fiber interactions, mechanical properties) and we will take it into account in the future. Unfortunately, the short time for the review of the article (only 3 days) does not allow us now to make this change. We will take into account for the following articles.
  2. We have revised all the language errors, as you indicated.

All changes are highlighted in the manuscript with red color.